# OPENING THE VOCABULARY OF NEURAL LANGUAGE MODELS WITH CHARACTER-LEVEL WORD REPRESENTATIONS

**Matthieu Labeau**
LIMSI-CNRS / Orsay, France
labeau@limsi.fr

**Alexandre Allauzen**
LIMSI-CNRS / Orsay, France
allauzen@limsi.fr

## ABSTRACT

This paper introduces an architecture for an open-vocabulary neural language model. Word representations are computed on-the-fly by a convolution network followed by pooling layer. This allows the model to consider any word, in the context or for the prediction. The training objective is derived from the Noise-Contrastive Estimation to circumvent the lack of vocabulary. We test the ability of our model to build representations of unknown words on the MT task of IWSLT-2016 from English to Czech, in a reranking setting. Experimental results show promising results, with a gain up to 0.7 BLEU point. They also emphasize the difficulty and instability when training such models with character-based representations for the predicted words.

## 1 INTRODUCTION

Most of neural language models, such as $n$-gram models Bengio et al. (2003) are word based and rely on the definition of a finite vocabulary $\mathcal{V}$. As a consequence, a Look-up table is associated to $\mathcal{V}$ in which each word $w \in \mathcal{V}$ is mapped to a vector of $d_E$ real valued features stored in a matrix $\mathbf{L} \in \mathbb{R}^{|\mathcal{V}|*d_E}$. While this approach has proven successful for a variety of tasks and languages, see for instance Schwenk (2007) in speech recognition and Le et al. (2012); Devlin et al. (2014); Bahdanau et al. (2014) in machine translation, it induces several limitations.

For morphologically-rich languages, like Czech or German, the lexical coverage is still an important issue, since there is a combinatorial explosion of word forms, most of which are hardly observed on training data. On the one hand, growing the Look-up table is not a solution, since it would increase the number of parameters without having enough training example for a proper estimation. On the other hand, rare words can be replaced by a special token. Nevertheless, this acts as a word class merging very different words without any distinction and using different word classes to handle out-of-vocabulary words Allauzen & Gauvain (2005) does not really solve this issue, since rare words are difficult to classify.

Moreover, for most inflected or agglutinative forms, as well as for compound words, the word structure is overlooked, wasting parameters for modeling forms that could be more efficiently handled by word decomposition. While the use of subword units Botha & Blunsom (2014); Sennrich et al. (2016) could improve the generalization power of such models, it relies on a proper and efficient method to induce these subword units.

To overcome these issues, we propose to investigate a word based language model with an open vocabulary. Since most of existing models and training criteria rely on the assumption of a finite vocabulary, the definition of an open vocabulary model, along with a training criterion, constitutes a scientific challenge. Our goal is to build word representations every words. Word representations are inferred on-the-fly from its character sequence, using convolution filters which implicitly capture subword patterns, as described in section 2. The architecture is based on a neural ngram model inspired from Bengio et al. (2003), while this idea can be extended to other kind of models. By relaxing the normalized constraint, the objective function borrows from the noise contrastive estimation Gutmann & Hyvärinen (2012) to allow our model to consider a possibly infinite vocabulary. This paper focusses on this challenge and its related training issues. To assess the efficiency of

this approach, the experimental setup described in section 3 uses a large scale translation task in a reranking setting. The experimental results summarized in section 4 show promising results as well as training issues.

## 2 MODEL DESCRIPTION

Word embeddings are parameters, stored in a Look-up matrix $\mathbf{L}$. The embedding $\mathbf{e}_w^{word}$ of a word $w$ is simply the column of $\mathbf{L}$ corresponding to its index in the vocabulary:

$$\mathbf{e}_w^{word} = [\mathbf{L}]_w$$

### 2.1 CHARACTER-LEVEL WORD EMBEDDINGS

To infer a word embedding from its character embeddings, we use a *convolution layer* Waibel et al. (1990); Collobert et al. (2011), similar to layers used in Santos & Zadrozny (2014); Kim et al. (2015). As illustrated in figure 1, a word $w$ is a character sequence $\{c_1, .., c_{|w|}\}$ represented by their embeddings $\{\mathbf{C}_{c_1}, .., \mathbf{C}_{c_{|w|}}\}$, where $\mathbf{C}_{c_i}$ denotes the vector associated to the character $c_i$. A convolution filter $\mathbf{W}^{conv} \in \mathbb{R}^{d_e} \times \mathbb{R}^{d_c * n_c}$ is applied over a sliding window of $n_c$ characters, producing local features :

$$x_n = \mathbf{W}^{conv}(\mathbf{C}_{c_{n-n_c+1}} : .. : \mathbf{C}_{c_n})^T + \mathbf{b}^{conv}$$

where $x_n$ is a vector of size $d_e$ obtained for each position $n$ in the word[1]. The notation $(\mathbf{C}_{c_{n-1}} : \mathbf{C}_{c_n})$ denotes the concatenation of two embeddings. The $i$-th element of the embedding of $w$ is the mean over the $i$-th elements of the feature vectors, passed by the activation function $\phi$ :

$$[\mathbf{e}^{char}]_i = \phi \left( \sum_{n=1}^{|w|-n_c+1} \frac{[\mathbf{x}_n]_i}{|w| - n_c + 1} \right) \tag{1}$$

Using a mean after a sliding convolution window ensures that the embedding combines local features from the whole word, and that the gradient is redistributed at scale for each character n-gram. The parameters of the layer are the matrices $\mathbf{C}$ and $\mathbf{W}^{conv}$ and the bias $\mathbf{b}^{conv}$.

### 2.2 MODELS

Our model follows the classic n-gram feedforward architecture. The input of the network is a $n$-words context $H_i = (w_{i-1}, \ldots, w_{N-i+1})$, and its output the probability $P(w|H_i)$ for each word $w \in \mathcal{V}$. The embeddings of the word in the context are concatenated and fed into a hidden layer:

$$\mathbf{h}^{H_i} = \phi(\mathbf{W}^{hidden}(\mathbf{e}_{i-1} : \ldots : \mathbf{e}_{N-i+1}) + \mathbf{b}^{hidden})$$

A second hidden layer my be added. Finally, the output layer computes scores for each word:

$$\mathbf{s}^{H_i} = \exp\left(\mathbf{W}^{out}\mathbf{h}^{H_i} + \mathbf{b}^{out}\right)$$

$\mathbf{W}^{hidden}$, $\mathbf{b}^{hidden}$, $\mathbf{W}^{out}$ and $\mathbf{b}^{out}$ are the parameters of the model. As the input Lookup-matrix $\mathbf{L}$, the output weight matrix $\mathbf{W}^{out}$ contains word embeddings, that are output representations of the words in the vocabulary:

$$\mathbf{e}_w^{out} = [\mathbf{W}^{out}]_w$$

Then, the output probabilities are expressed as:

$$P(w|H_i) = \frac{\exp \mathbf{e}_w^{out}\mathbf{h}^{H_i}}{\sum_{1<j<|\mathcal{V}|} \exp \mathbf{e}_j^{out}\mathbf{h}^{H_i}}$$

Later, we will use three different input layer to obtain word representations:

---

[1]Two padding character tokens are used to deal with border effects. The first is added at the beginning and the second at the end of the word, as many times as it is necessary to obtain the same number of windows than the length of the word. Their embeddings are added to $\mathbf{C}$.

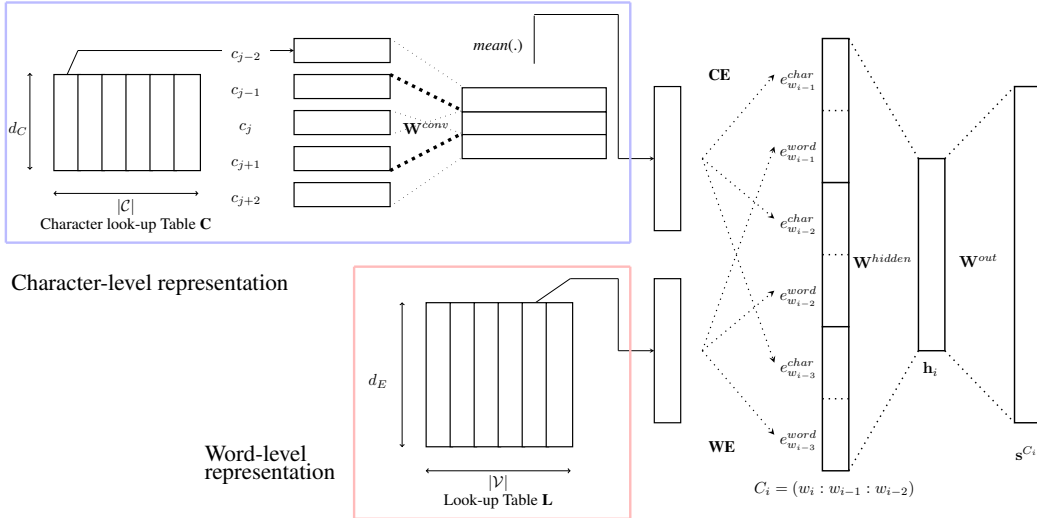

Figure 1: **CWE** Model architecture

- A classic NLM using word-level embeddings only, that we will note **WE**, which uses $|\mathcal{V}| * d_e$ parameters.
- A NLM using embeddings constructed from character n-grams by convolution + pooling, that we will note **CE**, which uses $|\mathcal{V}_c| * d_c + d_c * n_c * d_e$ parameters.
- A NLM using a concatenation of these two types of embeddings as word representation, that we will note **CWE**.

## 2.3 OBJECTIVE FUNCTION FOR OPEN VOCABULARY MODELS

Usually, such a model is trained by maximizing the log-likelihood. For a given word given its context, the model parameters $\theta$ are estimated in order to maximize the following function for all the n-grams observed in the training data:

$$LL(\theta) = \sum_{1 < i < |\mathcal{D}|} \log P_\theta(w_i | H_i).$$

This objective function raises two important issues. For conventional word models, it implies a very costly summation imposed by the softmax activation of the output layer. More importantly, this objective requires the definition of a finite vocabulary, while the proposed model may use character-based word embeddings, especially at the output, making the notion of vocabulary obsolete.

Therefore, the parameters estimation relies on Noise Contrastive Estimation (NCE) introduced in Gutmann & Hyvärinen (2012); Mnih & Teh (2012). This criterion allows us to train both types of models based on conventional word embeddings, along with character-based embeddings. The NCE objective function aims to discriminate between examples sampled from the real data and from a noise distribution. When presented with examples coming from a mixture of one sample from the data distribution $P_d$ and $k$ from the noise distribution $P_n$, $P^H(w \in \mathcal{D})$ denotes the posterior probability of a word $w$ given its context $H$ to be sampled from the training data $\mathcal{D}$. This probability can be expressed as follows:

$$P^H(w \in \mathcal{D}) = \frac{P_d^H(w)}{P_d^H(w) + kP_n(w)}$$

As suggested in Mnih & Teh (2012), $P_n$ only depends on $w$ here, since we chose the unigram distribution estimated on the training data. If

$$s_\theta^H(w) = \exp\left(\mathbf{e}^{out}\mathbf{h}^H + \mathbf{b}^{out}\right) \tag{2}$$

denotes the non-normalized score given by the model to a specific word $w$, as a function of the parameters $\theta$ and the context $H$, the final NCE objective function has the following form Gutmann

& Hyvärinen (2012):

$$J_\theta^H = E_{s_\theta^H} \left[ log \frac{s_\theta^H(w)}{s^H(w) + kP_n(w)} \right] + kE_{P_n} \left[ log \frac{kP_n(w)}{s_\theta^H(w) + kP_n(w)} \right],$$

where $s_\theta^H$ will tend to $P_d^H$ without the need for an explicit normalization.

## 2.4 CHARACTER-BASED OUTPUT WEIGHTS WITH NOISE-CONTRASTIVE ESTIMATION

The output weights representing each word in the vocabulary $\mathbf{e}^{out}$ can also be replaced by embeddings computed by a convolution layer on character $n$-grams. In this case the model can efficiently represent and infer a score to any word, observed during the training process or not, while with conventional word embeddings, out of vocabulary words only share the same representation and distribution. Instead of using a parameter matrix $\mathbf{W}^{out}$ to estimate the score like in equation 2, the output representation of a word $w$, $\mathbf{e}_w^{out}$ can be replaced by a vector $\mathbf{e}_w^{char-out}$ estimated on the fly based on its character sequence as described in equation 1, using $|\mathcal{V}_c| * d_c + d_c * n_c * d_h$ parameters. With this extension the model does not rely on a vocabulary anymore, hence motivating our choice of the NCE. This unnormalized objective allows us to handle an open vocabulary, since we only need to compute $k + 1$ word representations for each training examples. Models that use character-based embeddings both for input and output words are denoted by **CWE-CWE**.

Moreover, with this extension, the representations of words sharing character $n$-grams are tied. This is an important property to let the model generalize to unseen words. However, it can be also an issue: the limited number of updates for output representations ($k + 1$ words) has a "rich get richer" effect: the most frequent words are usually short and will get most of the update. They may therefore "contaminate" the representation of longer words with which they share character $n$-grams, even if these words are not related. This issue is further addressed in section 4.1.

## 3 EXPERIMENTAL SET-UP

The impact of the models described in section 2 is evaluated within the machine translation (MT) shared task of IWSLT-2016[2] from Englih to Czech. This language pair is highly challenging since Czech is a morphologically-rich language. Neural language models are integrated in a two steps approach: the first step uses a conventional MT system to produce an $n$-best list (the $n$ most likely translations); in the second step, these hypothesis are re-ranked by adding the score of the neural language model. To better benefit from the open vocabulary models introduced in section 2.1, a more complex system is also used: first an MT system is used to translate from English to a simplified form of Czech which is reinflected. With this pipeline we expect $n$-best lists with more diversity and also words unseen during the training process. The neural language models are then used to re-rank the reinflected $n$-best lists.

### 3.1 DATA

The IWSLT16 MT task is focused on the translation of TED talks. The translation systems are trained on parallel data from the **TED**, **QED** and **europarl**. Our Neural language models are trained on the same data, but training examples are sampled from these corpora given weights that are computed to balance between in-domain parallel data (**TED**), out-of domain parallel data, and additional monolingual data. Finally, we use the concatenation of **TED.dev2010**, **TED.dev2011** and **TED.tst2010** as development set, while **TED.tst2012** and **TED.tst2013** provide the test set.

### 3.2 CZECH RE-INFLECTION

In Czech, a morphologically rich language, each lemma can take a lot of possible word forms. Most of them won't appear - or with a very low frequency - in training data. For an important part of the words found in test data and unseen during training, their lemmas however can be observed but with a different morphological derivation.

---

[2]http://workshop2016.iwslt.org

A non-observed word form can't be generated by the translation system, and one seen too rarely won't be used in a relevant way. To circumvent this limitation, in a similar fashion as the method described in Marie et al. (2015), each noun, pronoun and adjective is replaced in the training corpora by its lemma along with some morphological features. These word forms are considered in factored way, where some of the POS tags are discarded to reduce the vocabulary. After the translation process, a cascade of Conditional Random Fields (CRF) are used to reintroduce the discarded features, such as gender, number and case, and to generate a new word form.

Formally, the MT system translates English into a simplified version of Czech, that is reinflected. Within this process, the MT system can produce a $n$-best list, that can be extended to a $nk$-best list, considering for each translation hypothesis the $k$-best reinflected sentences given by the factorized CRF. Intuitively, this process can introduce word forms potentially not yet seen in training data, but based on known paradigms, which can give an advantage to language models able to build a word representation from character $n$-grams.

### 3.3 BASELINE TRANSLATION SYSTEM

Our baseline is built with a Statistical Machine Translation system based on bilingual n-grams, NCODE[3], described in Crego et al. (2011). We follow the same setup as in Marie et al. (2015).

### 3.4 NLM TRAINING AND OPTIMIZATION

First, some comparative experiments on a smaller dataset are carried out to better understand how open vocabulary NLM behave and to set the hyper-parameters. First trained using stochastic gradient descent, we observed a quite unstable training process, restricting a proper hyper-parameters choices. We found that especially the embedding dimensions, and the activation functions used could make the NCE-objective hard to optimize. This was aggravated in Czech, which we found more difficult to work with than other morphologically complex languages, like German and Russian. The use of Adagrad Duchi et al. (2010) clearly helps to solve most of these issues, but adds consequent computation time. Following preliminary results on our work with a similar model on a different task Labeau et al. (2015), we made the choice of not implementing LSTMs to obtain character-level word representations. It gave similar results, at the cost of unstable training and extended computation time. We then train using batches of 128, for various context sizes, **WE**, **CWE**, and **CWE-CWE** models. The ReLu activation function is used, along with an embedding size of $d_e = 128$. When relevant, we used a character embedding size of $d_c = 32$ and a convolution on $n_c = 5$-grams of characters for all experiments[4]. Concerning the NCE training, we sampled $k = 25$ examples from the unigram distribution obtained from the training data, for each example sampled from the data. The models were implemented using C++[5].

### 3.5 RERANKING

The re-ranking step uses additional features to find a better translation among the $n$-best generated by the decoder (in our case, $n = 300$): we use the score (probability) of **WE**, **CWE** and **CWE-CWE** models given to each sentence by our models as such a feature. Tuning for re-ranking was performed with KB-MIRA Cherry & Foster (2012), and evaluation using BLEU score.

## 4 EXPERIMENTAL RESULTS

The first set of experiments investigates the impact of the padding design on the character-level representation followed by a study of the learning behavior of our proposed models and training criterion. Then, the proposed models are evaluated within the MT task. The final set of experiments analyzes the issues of the model based on character-level representation for output words, in order to propose remedies.

---

[3]http://ncode.limsi.fr

[4]Results did not differ significantly when increasing these embedding sizes, with an impact on convergence speed and computation time.

[5]Implementation will be made available.

## 4.1 TIES BETWEEN CHARACTER-LEVEL REPRESENTATION OF OUTPUT WORDS

Preliminary results on smaller dataset are quite poor for models using character-level representation, and far worse when used for the output layer. We suspect that groups of characters are updated far more together, yielding a "contamination" of several character n-grams by very frequent short words.

Indeed, our simple padding scheme, as shown in the left part of table 1, makes words sharing first or last letter(s) systematically share at least one character n-gram: we suppose it gives the models more chance to detect similarities in word forms sharing prefixes and suffixes.

The representations of any of the character n-grams that are included in the frequent words will thus be re-used in a large part of the other words in the corpus. A huge number of word forms are affected: a little more than one third of the training data shares its first character n-gram with one of the ten most frequent words, and a little more than one quarter shares its last.

While considering varying size of character n-grams when building our word representation, as in Kim et al. (2015), would certainly help, it would increase our computation time. We thus choose to alleviate our padding scheme, as shown on the right part of table 1. We add only one character token at the beginning of the word, and one at the end[6]. While it may inhibit the capacity of the model to build links between words sharing prefixes or suffixes, it improves results drastically, especially when using character-level outputs, as shown in figure 3. This limited padding scheme is used for the following experiments.

| | | | |
|---|---|---|---|
| ○ ○ ○ ○ $a$ ● ● ● ● | | ○ ○ $a$ ● ● | |
| ○ ○ ○ ○ $a\,l\,e$ ● ● ● ● | ○ ○ ○ ○ $n\,a$ ● ● ● ● | ○ $a\,l\,e$ ● | ○ ○ $n\,a$ ● |
| ○ ○ ○ ○ $a\,b\,y$ ● ● ● ● | ○ ○ ○ ○ $z\,a$ ● ● ● ● | ○ $a\,b\,y$ ● | ○ ○ $z\,a$ ● |
| ○ ○ ○ ○ $a\,\check{z}$ ● ● ● ● | ○ ○ ○ ○ $b\,y\,l\,a$ ● ● ● ● | ○ ○ $a\,\check{z}$ ● | ○ $b\,y\,l\,a$ ● |
| ○ ○ ○ ○ $a\,n\,i$ ● ● ● ● | ○ ○ ○ ○ $d\,v\,a$ ● ● ● ● | ○ $a\,n\,i$ ● | ○ $d\,v\,a$ ● |
| ○ ○ ○ ○ $a\,s\,i$ ● ● ● ● | ○ ○ ○ ○ $t\,\check{r}\,e\,b\,a$ ● ● ● ● | ○ $a\,s\,i$ ● | ○ $t\,\check{r}\,e\,b\,a$ ● |

Table 1: Padding for word decomposition in character 5-grams: ○ is a character token indicating the beginning of the word, while ● indicates the end of the word. The left part of the table shows our original padding scheme, which makes very different words share character 5-grams, especially with short, frequent words. The right part of the table shows our alleviated padding scheme.

## 4.2 NLM TRAINING

While the perplexity of our language models is not our main focus, it is still related to the quantity that our training seeks to optimize - since the NCE gradient approaches the maximum likelihood gradient Mnih & Teh (2012). On figure 2 are shown perplexity values of each model during training. These values are based on a vocabulary containing the 250K most frequent words on the training data - it is also the vocabulary used in the model when relevant. They are computed on the development set after each epoch. An epoch includes 2,5M N-grams sampled from the training data. On table 2 are shown the best perplexity obtained on the development set by each model, during training.

| Context size (Number of words) | 3 | 6 |
|---|---|---|
| WE | 227 | 193 |
| CWE | 207 | 185 |
| CWE-CWE | 308 | 243 |

Table 2: Best perplexity reached on the development set, on a 250K output vocabulary, after 15 epochs of 2,5M n-grams

Table 2 shows that a character-level word representation helps to decrease the perplexity, even if a larger context closes the gap. To compute the perplexity of **CWE-CWE** models, we use the

---

[6]For short words, we add the numbers of tokens necessary for the word to have at least $n_C = 5$ characters, as shown in table 1

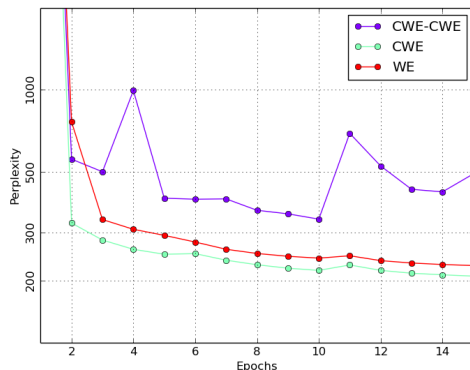 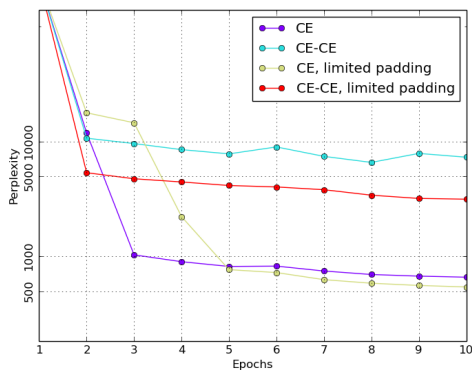

Figure 2 Figure 3

Figure 4: Model perplexity measured on the development set during training. The context size is 3 words. Figure 3 shows models based on character-level word representations, with and without complete padding. Models are trained on the same data than Figure 2 but on smaller epochs (250K n-grams).

same vocabulary as for other models, and use the 'unknown' tokens for words and characters-based representations. Hence, the perplexity computed is difficult to interpret. The main downside of Adagrad is that the learning rate determined by accumulating the history of past gradients is usually too aggressive and stops learning rather early. We simply reset this history every five epochs to give the model a chance to improve, which explains the flattening followed by small improvements we see for **WE** and **CWE** models. We choose to do that reset 2 times, based on previous experiments. Despite adaptive gradient, training of **CWE-CWE** models stays unstable.

### 4.3 RERANKING

| | System to be re-ranked | BLEU Reference | CWE | | CWE-CWE | | WE | |
|---|---|---|---|---|---|---|---|---|
| | | | n=3 | n=6 | n=3 | n=6 | n=3 | n=6 |
| En $\to$ Cz | Baseline system | 19.6 | 20.1 | **20.3** | 19.8 | 20.0 | 20.0 | 20.2 |
| En $\to$ Simplified Cz | Reinflected baseline system | 19.5 | 20.0 | 20.2 | 19.6 | 20.1 | 20.1 | 20.0 |
| | 3-best Reinflected baseline system | | 19.9 | **20.3** | 19.6 | 20.0 | 20.1 | 20.1 |
| | 5-best Reinflected baseline system | | 19.9 | **20.3** | 19.5 | 19.9 | 20.0 | 20.1 |

Table 3: Best BLEU score obtained after n-best reranking of the hypothesis given by the translation and translation + k-best reinflection systems. $n$ is the context size (in number of words)

The reranking results are shown in table 3. The first line corresponds to experiments with a direct translation from English to Czech, where $n$-best lists generated by the MT system are simply rescored by our models. The best result is given by the longest-context **CWE** model, which produces a $+\mathbf{0.7}$ BLEU score improvement. **CWE** models gives on average $+0.1$ BLEU point compared to **WE** models, while **CWE-CWE** are $-0.2$ BLEU point under. Doubling the context size consistently improves results of $+0.2$ BLEU point.

Experimental results on reinflected Czech seems to follow a similar trend: **CWE** models behave a little better than **WE** models, while **CWE-CWE** models are under. While simply reranking $n$-best lists is not as efficient as doing it directly in Czech, reranking $nk$-best lists extended by the factorized CRF gives a small improvement, reaching an improvement of $+\mathbf{0.7}$ BLEU point. As a general rule, small context models seem to have difficulties with reinflected Czech. The main advantage given by the **CWE** model is an ability to better rerank $nk$-best lists. These results suggest that, while the normalization + reinflection procedure may introduce diversity in the output to be reranked, our models are not able to draw any significant advantage from it.

### 4.4 ANALYSIS OF CHARACTER-LEVEL OUTPUT REPRESENTATIONS PERFORMANCE

Models using character-level output representations gave sub-par results on re-ranking. It is surprising, especially for re-inflected Czech: such a model is supposed to behave better on unknown words, and thus should benefit from diversity given by generating new words. However, as we can see in table 4, re-inflection doesn't add that much diversity (About 0.1 % of OOV words, and about 0.001 % of words never seen by the model before). Diversity is also inhibited by our training algorithm: while we train open-vocabulary models, the negative examples used with Noise-contrastive estimation come from a closed vocabulary.

|  | Full training vocabulary | 250K words vocabulary |
|---|---|---|
| Reference | 0.131 % | 0.995 % |
| En → Cz (300-best) | 0.566 % | 1.173 % |
| En → Simplified Cz + Reinflection | 0.567 % | 1.263 % |
| En → Simplified Cz + 3-Best reinflection | 0.567 % | 1.277 % |
| En → Simplified Cz + 5-Best reinflection | 0.568 % | 1.285 % |

Table 4: Ratio of unknown words in system outputs measured on the test set.

This can related to the nature of the unigram distribution used to sample negative examples. As explained in section 4.1, it makes frequent short words completely outweigh the others in number of updates, and we are forced to reduce the ability of the model to find common morphological attributes between words to avoid 'contamination' of character n-gram representations.

## 5 RELATED WORKS

There is a number of different strategies to efficiently train NNLMs with large vocabularies, such as different types of hierarchical softmax Mnih & Hinton (2009); Le et al. (2011), importance sampling Bengio & Sénécal (2003), and Noise contrastive estimation Gutmann & Hyvärinen (2012); Mnih & Teh (2012). Vaswani et al. (2013) has showed the interest of training a NLM with NCE to re-rank $k$-best lists, while Devlin et al. (2014) uses a self-normalization. Recently, a comparative study Chen et al. (2016) has been made on how to deal with a large vocabulary. However, the purpose of this paper is to explore models with open vocabulary rather large vocabulary.

There is a surge of interest into using character-level information for a wide range of NLP tasks, with improved results in POS Tagging Santos & Zadrozny (2014), Text classification Zhang & LeCun (2015), Parsing Ballesteros et al. (2015), Named entity recognition Lample et al. (2016).

In language modeling, first applications to language modeling were strictly using characters, and performed less than word-level models Mikolov et al. (2012), while showing impressive results for text generation Sutskever et al. (2011); Graves (2013), using bi-directional LSTM Graves et al. (2013). Recently, Ling et al. (2015) has used bi-directional LSTM to build word representations from characters, with improvements in language modeling and POS-tagging.

The recent work of Kim et al. (2015), that uses convolutional networks and pooling to construct a word representation from character n-grams, coupled with highway networks Srivastava et al. (2015), showed on various languages that using characters improves results on the language modeling task (for a small corpus), even more so for languages with complex morphology. A similar architecture was used Józefowicz et al. (2016) on a larger dataset, conjointly with bi-directional LSTMs, and trained with importance sampling, showing great results.

On the study of NNLMs in the context of Machine Translation, we can mention the work of Luong et al. (2015) on the effect of the number of layers on reranking $n$-best lists. Finally, while not directly related to our work, Luong & Manning (2016) very recently showed great improvements on a translation task by handling rare words with character-level recurrent networks, with a neural translation model.

## 6 CONCLUSION

In this work, we addressed the challenge of designing an open vocabulary Neural Language Model. For that purpose, word representations are estimated on-the-fly from n-grams of characters. Two kinds of models are introduced: first, NLMs using word and character-level embeddings to represent the input context (**CWE**); then its extension to an open-vocabulary even for the predicted words (**CWE-CWE**). These models were used to re-rank outputs of translation systems from English to Czech. We also carried out experiments on translation systems from English to a simplified Czech, which is then re-inflected into Czech before re-ranking.

We obtained a slight improvement in BLEU score using a **CWE** model, which, given the little variety of the words generated by translation systems, makes us suppose there is room for more. We plan to investigate with more complex translation systems, as well as with other applications, such as morphological re-inflection.

While the performance of our open-vocabulary models are to some extent disappointing, they open questions about the learned representations we will explore. We also plan to investigate on a more fitted noise distribution to use with NCE when training open-vocabulary models.

ACKNOWLEDGMENTS

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
