# Peer review of "Opening the vocabulary of  neural language models with character-level word representations"

_ICLR 2017 — rejected_

[Reviewer Comment · AnonReviewer2 · 01 Dec 2016]
**PPL for open vocabulary**

In Sec. 4.2 you mention that perplexity is hard to interpret for models not using an explicit output vocabulary. When analysing open vocabulary approaches, perplexity can also be renormalized to character level, cf. e.g. Shaik et al. IWLST 2013. Did you consider this?

[Reviewer Comment · AnonReviewer2 · 01 Dec 2016]
**Results using character-level representations**

Also Kim et al. AAAI 2015 got the similar conclusions w.r.t. the performance of character-level embeddings and also provided a discussion with suggestions for improvements. Did you consider these?

[Reviewer Comment · AnonReviewer2 · 01 Dec 2016]
**NCE tuning**

Can you provide more details on the configuration of the NCE training?

[Reviewer Comment · AnonReviewer2 · 01 Dec 2016]
**Use of  feed-forward NN**

From your notation I get that you used a feed-forward NN, can you confirm?

[Reviewer Comment · AnonReviewer2 · 01 Dec 2016]
**Sec. 2.1: Character-level word embedding**

Can you confirm that the character-level word embedding used here is the same as in the google paper by Kim et al. AAAI 2015? It is not cited in Sec. 2.1.

[Reviewer Comment · AnonReviewer2 · 01 Dec 2016]
**Notation**

Pls. define the use of the colon in the first equation of Sec. 2.1

P^H((w:H)\in D) is not defined before the second equation in Sec. 2.3. Also, in the sentence introducing this equation to refer to "this probability" - please provide an explicit reference to what probability is meant here.

Pls. define e^{out} and e^{char-out} in Sec. 2.4 - are they the same as e^{out}_w in Sec. 2.2 and e^{char} in Sec. 2.1?

[Official Review · AnonReviewer2 · rating 4 · confidence 4 · 15 Dec 2016]

In this submission, an interesting approach to character-based language modeling is pursued that retains word-level representations both in the context, and optionally also in the output. However, the approach is not new, cf. (Kim et al. 2015) as cited in the submission, as well as (Jozefowicz et al. 2016). Both Kim and Jozefowicz already go beyond this submission by applying the approach using RNNs/LSTMs. Also, Jozefowicz et al. provide a comparative discussion of different approaches to character-level modeling, which I am missing here, at least by discussing this existing work. THe remaining novelty of the approach then would be its application to machine translation, although it remains somewhat unclear, inhowfar reranking of N-best lists can handle the OOV problem - the translation-related part of the OVV problem should be elaborated here. That said, some of the claims of this submission seems somewhat exaggerated, like the statement in Sec. 2.3: "making the notion of vocabulary obsolete", whereas the authors e.g. express doubts concerning the interpretation of perplexity w/o an explicit output vocabulary. For example modeling of especially frequent word forms still can be expected to contribute, as shown in e.g. arXiv:1609.08144

Sec. 2.3: You claim that the objective requires a finite vocabulary. This statement only is correct if the units considered are limited to full word forms. However, using subwords and even individual characters, implicitly larger and even infinite vocabularies can be covered with the log-likelihood criterion. Even though this require a model different from the one proposed here, the corresponding statement should qualified in this respect.

The way character embeddings are used for the output should be clarified. The description in Sec. 2.4 is not explicit enough in my view.

Concerning the configuration of NCE, it would be desirable to get a better idea of how you arrived at your specific configuration and parameterization described in Sec. 3.4.

Sec. 4.1: you might want to mention that (Kim et al. 2015) came to similar conclusions w.r.t. the performance of using character embeddings at the output, and discuss the suggestions for possible improvements given therein.

Sec. 4.2: there are ways to calculate and interpret perplexity for unknown words, cf. (Shaik et al. IWSLT 2013).

Sec. 4.4 and Table 4: the size of the full training vocabulary should be provided here.

Minor comments:
p. 2, bottom: three different input layer -> three different input layers (plural)
Fig. 1: fonts within the figure are way too small
p. 3, first item below Fig. 1: that we will note WE -> that we will denote WE
Sec. 2.3: the parameters estimation -> the parameter estimation (or: the parameters' estimation)
p. 5, first paragraph: in factored way -> in a factored way
p. 5, second paragraph: a n-best list, a nk-best list -> an n-best list, an nk-best list
Sec. 4.2, last sentence: Despite adaptive gradient, -> verb and article missing

[Official Review · AnonReviewer3 · rating 2 · confidence 5 · 17 Dec 2016]
**lacks experimental evidence**

this paper proposes a model for representing unseen words in a neural language model. the proposed model achieves poor results in LM and a slight improvement over a baseline model. 

this work needs a more comprehensive analysis:
- there's no comparison with related work trying to address the same problem
- an intrinsic evaluation and investigation of why/how their work should be better are missing.
- to make a bolder claim, more investigation should be done with other morphologically rich languages. Especially for MT, in addition to going from En-> Language_X, MRL_X -> En or MRL_X -> MRL_Y should be done.

[Official Review · AnonReviewer1 · rating 3 · confidence 4 · 18 Dec 2016]

This paper proposes an extension of neural network language (NLM) models to better handle large vocabularies. The main idea is to obtain word embeddings by combining character-level embeddings with a convolutional network.

The authors compare word embeddings (WE),character embeddings (CE) as well a combined character and word embeddings (CWE). It's quite obvious how CE or CWE embeddings can be used at the input of an NLM, but this is more tricky at the output layer. The authors propose to use NCE to handle this problem.  NCE allows to speed-up training, but has no impact on inference during testing: the full softmax output layer must be calculated and normalized (which can be very costly).

It was not clear to me how the network is used during TESTING with an open-vocabulary. Since the NLM is only used during reranking, the unnormalized probability of the requested word could be obtained at the output. However, when reranking n-best lists with the NLM feature, different sentences are compared and I wonder whether this does work well without proper normalization.

In addition, the authors provide perplexities in Table 2 and Figures 2 and 3.  This needs normalization, but it is not clear to me how this was performed.  The authors mention a 250k output vocabulary. I doubt that the softmax was calculated over 250k values. Please explain.

The model is evaluated by reranking n-best lists of an SMT systems for the IWSLT 2016 EN/CZ task.  In the abstract, the authors mention a gain of 0.7 BLEU. I do not agree with this claim. A vanilla word-based NLM, i.e. a well-known model, achieves already a gain of 0.6 BLEU. Therefore, the new model proposed in this paper brings only an additional improvement of 0.1 BLEU. This is not statistically significant. I conjecture that a similar variation could be obtained by just training several models with different initializations, etc.

Unfortunately, the NLM models which use a character representation at the output do not work well. There are already several works which use some form of character-level representations at the input.

Could you please discuss the computational complexity during training and inference.

Minor comments
 - Figure 2 and 3 have the caption "Figure 4". This is misleading.
 - the format of the citations is unusual, eg.
   "While the use of subword units Botha & Blunsom (2014)"
   -> "While the use of subword units (Botha & Blunsom, 2014)"

[Final Decision · Program Chairs · 06 Feb 2017]
**ICLR committee final decision**

The reviewers raise several important questions about modeling and methodology that should be answered in later versions of the paper. The paper also overstates its findings.